# Perception regarding live kidney donation in the general population of South Korea

Eunjeong Kang[1], Jangwook Lee[2], Sehoon Park[3], Yaerim Kim[4], Hyo Jeong Kim[5], Yong Chul Kim[6], Dong Ki Kim[6], Kwon Wook Joo[6], Yon Su Kim[6], Insun Choi[5], Hajeong Lee[6]*

1 Department of Internal Medicine, Ewha Womans University Seoul Hospital, Ewha Womans University College of Medicine, Seoul, Korea, 2 Department of Internal Medicine, Dongguk University Ilsan Hospital, Goyang, Korea, 3 Department of Biomedical Sciences, Seoul National University College of Medicine, Seoul, Korea, 4 Department of Internal Medicine, Keimyung University School of Medicine, Daegu, Korea, 5 Division of Healthcare Technology Assessment Research, National Evidence-based Healthcare Collaborating Agency, Seoul, Korea, 6 Department of Internal Medicine, Seoul National University Hospital, Seoul, Korea

☯ These authors contributed equally to this work.
* mdhjlee@gmail.com

**Data Availability Statement:** All relevant data are within the manuscript and its Supporting information files.

**Funding:** This study was supported by the National Evidence Collaborating Agency (project number

## Abstract

This study aimed to know how the general population recognizes live kidney donation in Korea. Participants were randomly selected from the general population after proportional allocation by region, sex, and age. Selected participants received a questionnaire that included demographic information, socioeconomic and marital statuses, prior recognition of live donor kidney transplantation, expected changes after donation, and the need for support after donor nephrectomy. Among the 1,000 participants from the web-based survey, 83.8% answered they fully understood living donor kidney transplantation, 81.1% knew about them, and 51.1% were willing to donate. Various complications after nephrectomy and deterioration in health after donation were the most significant reasons for those reluctant to donate. Most agreed that the government should provide social and economic support to living kidney donors, especially after exposure to the description of donor nephrectomy. Financial support, including surgery and regular medical check-up costs, was the most preferred government support. The Korean general population seemed aware of the value and safety of kidney donation, although only half of them were willing to donate due to concerns about possible complications. Most participants agreed on social and economic support for living kidney donors, especially surgery-related costs.

## Introduction

Kidney transplantation is the best alternative treatment option for patients with end-stage kidney disease (ESKD). Considering superior graft and patient survival, live donor kidney transplants have been the preferred option over deceased donor kidney transplantations [1]. South Korea is one of the leading countries that depend mainly on living donors rather than on deceased donors for kidney transplantation [2–4]. According to the Organ

NECA-A-20-005, URL: https://www.neca.re.kr/eng/index.do) and Pf. Hajeong Lee received research fund. The funder had no role in performing the study, and the study was independently performed by the authors.

**Competing interests:** The authors have declared that no competing interests exist.

Procurement and Transplantation Network, the rate of living donor kidney transplants reached 27.63 cases per million, ranking second out of 70 countries in South Korea in 2020 [5].

Since the commitment to giving a kidney to others is a noble decision from a generous heart, it is essential to help donors maintain medical, psychological, and socioeconomic stability before and after donor nephrectomy. Recent studies on the medical safety of kidney donation, including all-cause mortality, ESKD, pregnancy-related complications, and development of hypertension, have raised concerns regarding the long-term prognosis of living kidney donors [1, 6, 7], although the absolute risks were not high. Besides, it is unclear whether the cause of death was related to donation since regular follow-up was not conducted with most donors. Some researchers also suggested that the risks of ESKD after donor nephrectomy were caused by the "first hit" presented at birth or a "second hit" acquired later in life [8]. Especially according to the results of a study in Korea, long-term mortality was not higher than that of matched healthy controls, and the study showed that non-medical factors, such as socioeconomic status and residence, had a significant impact on the long-term prognosis of living kidney donors [7, 9]. Therefore, it seems necessary to consider psychological or socioeconomic factors and medical factors for the safety of living kidney donors [10].

In Korea, live kidney donation does not have a systemic reimbursement system for donation-related medical costs, including money paid for the work-up, receiving surgery, hospitalization, visiting outpatient clinics, and out-of-pocket money or lost wages. Moreover, in some cases, individuals who have donated kidneys are restricted from signing up for private insurance or do not reimburse the costs of donor nephrectomy with private insurance. Additionally, according to a survey on live organ donors in Korea, donors often lost opportunities for promotion or were disadvantaged because of sick leave during the recovery after donor nephrectomy, even among many donors that failed to be employed. This situation was confirmed in a recent study using data from the National Health Insurance System database [11]. Namely, living kidney donors may have a higher rate of loss of employment and a lower rate of being newly employed in the short term after kidney donation, which results in deterioration of economic status compared with matched healthy individuals [12].

The present survey study investigated the perception and attitude of live donor kidney transplantations among the general population of South Korea. We believe that the general population could possibly be future living kidney donors. In addition, it is thought that a society-wide consensus is essential to determine whether a policy for expanding economic support to kidney donors is needed. Therefore, we also evaluated whether there were any changes in donor attitudes before and after providing an accurate and detailed description of live donor kidney transplants. This survey will serve as a basis for listening to social consensus on the current support of living kidney donors.

## Materials and methods

### Ethics statements

The study protocol was approved by the institutional review board of the Seoul National University Hospital (approval number: H-1903-116-1019) and the National Evidence-based Healthcare Collaborating Agency in 2019, and it is reported in accordance with the STROBE checklist. The study was conducted in accordance with the principles of the Declaration of Helsinki. Written informed consent was obtained from participants before they replied to the questionnaire.

## Recruitment

We conducted a web-based survey to assess the attitude of the general population toward live donor kidney transplants. Survey recruitment was conducted from February-May 2020. Eligible participants were required to be (1) aged more than 19 years and (2) a resident of South Korea. All respondents were asked to read a summary page explaining the purpose and content of the questionnaire prior to starting the survey. Respondents read the study description and then chose whether to participate in the study.

The sampling frame was developed by a research company (Hankook Research Inc., Seoul, Korea), which sent a questionnaire via email and collected the responses through a computer-assisted web interview. The participants were randomly anonymized and extracted after proportional allocation by region, sex, and age based on the resident registration population status in January 2020.

## Questionnaire design

Some contents of the questionnaire were extracted from the Korea National Health and Nutrition Examination Survey, and some parts of the questionnaire were developed by the study researchers in consultation with nephrologists [11, 13]. In the refinement of the survey questions and layout, we obtained feedback from two experienced transplantation coordinators on the comprehensibility, usability, and time taken to complete the survey from the perspective of the target participants. The questionnaire was written in Korean. Only completed questionnaires were submitted. The entire questionnaire was translated into English language and provided as S1 File.

## Measures

The questionnaire included demographic items regarding age, sex, residence, religion, educational attainment, marital status, underlying disease, and the number of family members. The contents of the survey also included prior recognition of live donor kidney transplantation, purpose and recipients of donation, expected changes after donation, and the need for support after donor nephrectomy. We asked all participants of this study to read a detailed explanation provided to living kidney donors before kidney donation; it contained the precise surgical procedures of donor nephrectomy, possible perioperative complications, and possible long-term adverse medical outcomes. After presenting the description to living kidney donors during the survey, the questionnaire survey was conducted with the same question to investigate whether there was any change in responses. All of the questionnaire contents can be found in the supplementary digital content.

## Statistical analysis

Descriptive statistics were used to summarize the survey responses. The chi-squared test was used to compare categorical variables between groups. For each questionnaire, a frequency analysis was performed, and McNemar's [14] and paired t-tests were used to determine whether there was a difference in responses before and after the description of kidney donation operation and before and after kidney donation. We examined Cronbach's alpha coefficient to estimate the internal consistency of the item, which was confirmed to be 0.810. All statistical analyses were performed using IBM SPSS Statistics software (version 23.0; IBM Corp., Armonk, NY, USA). Statistical significance was set at $p < 0.05$.

## Results

### Baseline characteristics of the respondents

In total, 1,000 participants completed the survey, and their baseline characteristics are summarized in Table 1. The median age of the study participants was 47±15.2 years, and 491 (49.1%) were men. Of the respondents, 88.9% lived in metropolitan or small-medium-sized cities, and 57.9% graduated from college or beyond. Overall, 71.3% had not been diagnosed with any chronic disease. The most significant number of respondents were non-religious (46.7%), followed by Christians (24.6%), Buddhists (15.9%), and Catholics (11.9%).

The survey was conducted with the same questions after exposing the description provided to the donor during transplantation consultations to check whether there were any changes in the responses. After reading the explanation regarding kidney transplants, 83.8% of respondents answered that they fully understood living donor kidney transplantation. Of those who answered an open-ended question asking if they had any further questions, 6.4% of participants were curious about the side effects after donation, and 4.4% asked about the reward and social support for donors.

### Awareness and perceptions of live donor kidney transplantation

In the initial survey, 811 (81.1%) respondents were aware of living kidney donation. In a question regarding their level of awareness of live donor kidney transplantation, more than half of the participants replied that they knew very well (4.6%) or to a certain extent (52.9%), while the other 42.5% said that they did not know well or had only heard of it (S1 Fig). Regarding the safety of kidney donation, 31.9% of the participants thought it was relatively safe, and 22.3% thought it was unsafe (Fig 1A). Many survey participants considered that kidney donation would affect donors' long-term health (82.6%, Fig 1B).

Interestingly, the participants tended to perceive live kidney donation as safer after reading a detailed description of living donor transplantation compared with their initial response (Fig 1A). Similarly, the proportion of respondents that believed kidney donation may not affect donors' health was increased, although the majority still worried about the adverse effect on the overall health status after kidney donation (Fig 1B).

### Individual willingness to donate a kidney

When asked if they were willing to donate their kidneys, 511 (51.1%) answered positively. The distribution of religion, marital status, and chronic disease status did not differ significantly depending on the willingness to donate a kidney (Table 2). When multiple choices were allowed, participants wanted to donate their kidneys to of their offspring (86.3%), spouse (85.5%), sibling (80.2%), parent (75.9%), a close friend (31.9%), relative (19.0%), and others (6.8%).

Among the 511 participants willing to donate a kidney, the most common reason to donate a kidney was philanthropic causes, including self-satisfaction after saving someone they love (69.7%) and promoting the recipient's health (67.7%). Only 19% of them considered kidney donation to have no possible adverse health effects (Fig 2A).

Among the 136 participants who were not willing to donate a kidney, the most common reasons for reluctance to donate a kidney were possible complications driven by surgical procedures and hospitalization that were never needed for their own sake in substance. They reported fear of adverse physical complications after nephrectomy (69.1%) and long-term adverse medical outcomes (54.4%). Additionally, 33.8% of the respondents answered that they were afraid of affecting their economic status after the donation (Fig 2B). There was no

**Table 1. Baseline characteristics of survey participants.**

| Variables | Data (n = 1,000) |
|---|---|
| **Sex** | |
| Male | 491 (49.1) |
| Female | 509 (50.9) |
| **Age (in years)** | |
| $\geq$ 20 and < 30 | 167 (16.7) |
| $\geq$ 30 and < 40 | 161 (16.1) |
| $\geq$ 40 and < 50 | 193 (19.3) |
| $\geq$ 50 and < 60 | 199 (19.9) |
| $\geq$ 60 | 280 (28.0) |
| **Regional scale** | |
| Metropolitan | 441 (44.1) |
| Small-medium sized cities | 448 (44.8) |
| Town | 111 (11.1) |
| **Educational attainment** | |
| High school graduates and below | 421 (42.1) |
| College or beyond | 579 (57.9) |
| **Job** | |
| Agriculture/forestry/fishing | 11 (1.1) |
| Self-employment | 59 (5.9) |
| Sales and service | 91 (9.1) |
| Production and labor | 98 (9.8) |
| Management/professional | 355 (35.5) |
| Housewives | 157 (15.7) |
| Students | 62 (6.2) |
| Unemployed/retired/others | 167 (16.7) |
| **Household monthly income** | |
| Less than $1,600 | 193 (19.3) |
| $1,600 to < $3,200 | 397 (39.7) |
| $3,200 to $4,800 | 227 (22.7) |
| More than $4,800 | 151 (15.1) |
| Unknown | 32 (3.2) |
| **Religions** | |
| Buddhism | 159 (15.9) |
| Christianity | 246 (24.6) |
| Catholicism | 119 (11.9) |
| Others | 9 (0.09) |
| None | 467 (46.7) |
| **Presence of chronic disease** | |
| Yes | 287 (28.7) |
| No | 713 (71.3) |
| **National Health Insurance System** | |
| National Health Insurance (Community) | 279 (27.9) |
| National Health Insurance (Workplace) | 618 (61.8) |
| Full-aided | 19 (1.9) |
| Partial-aided | 6 (0.6) |
| Unsubscribed/unknown | 77 (7.7) |

Data are presented as number (%) for categorical variables.

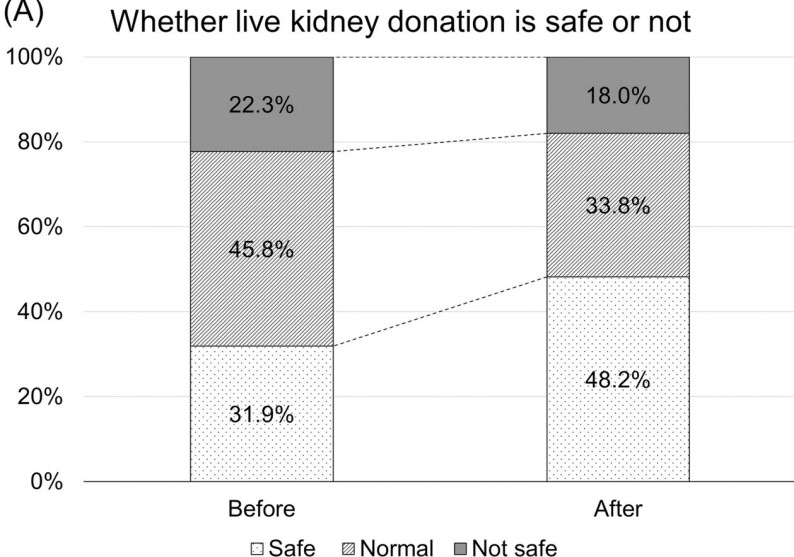

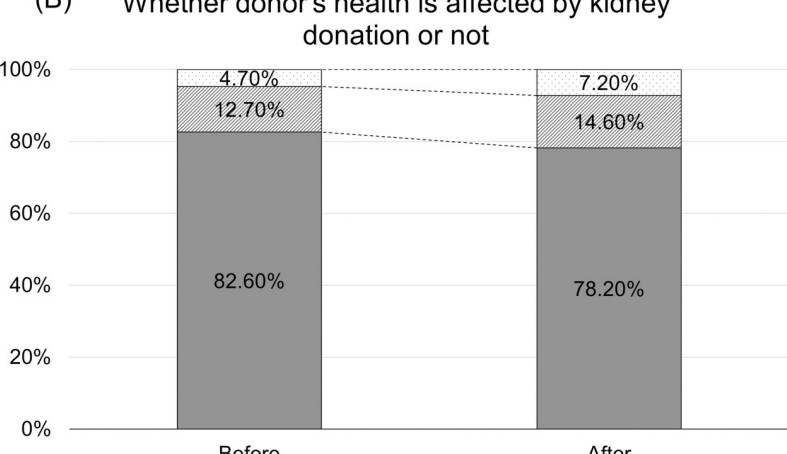

**Fig 1. Changes in perception of transplants before and after reading descriptions of living donor kidney transplants.** (A) Whether kidney donation is safe. (B) Whether donor's health is affected by kidney donation of not.

statistically significant difference in the willingness to donate kidneys after providing detailed explanations regarding kidney transplantation (p = 0.076).

### Opinions on social support for living kidney donors

Initially, most participants were more likely to agree that the government should provide social and economic support to living kidney donors (yes, 73.2%; no, 8.3%; unsure, 18.5% of the total 1,000 participants). When asked what kind of support should be provided to the donors by allowing duplicate responses, the results were in the following order: support for nephrectomy and hospitalization expenses (74.2%), support of hospital expenses for follow-up monitoring of renal function after donor nephrectomy (70.1%), justification of sick leave within a certain

**Table 2. Characteristics according to willingness of kidney donation[a].**

| Characteristics | Willing to donate | Do not intend to donate | Undetermined | Total | p |
|---|---|---|---|---|---|
| | (N = 511) | (N = 136) | (N = 353) | (N = 1000) | |
| **Marital status** | | | | | 0.75 |
| Married | 370 (72.4) | 94 (69.1) | 254 (72.0) | 718 (71.8) | |
| Unmarried | 141 (27.6) | 42 (30.9) | 99 (28.0) | 282 (28.2) | |
| **Religion** | | | | | 0.13 |
| Buddhism | 78 (15.3) | 19 (14.0) | 62 (17.6) | 159 (15.9) | |
| Christianity | 134 (26.2) | 34 (25.0) | 78 (22.1) | 246 (24.6) | |
| Christianity | 65 (12.7) | 13 (9.6) | 41 (11.6) | 119 (11.9) | |
| Others | 1 (0.2) | 2 (1.5) | 6 (1.7) | 9 (0.9) | |
| None | 233 (45.6) | 68 (50.0) | 166 (47.0) | 467 (46.7) | |
| **Dedication in religious life** | | | | | 0.66 |
| Dedicate very hard | 29 (10.4) | 7 (10.3) | 16 (8.6) | 52 (9.8) | |
| Dedicate hard | 101 (36.3) | 23 (33.8) | 60 (32.1) | 184 (34.5) | |
| Pretty well | 98 (35.3) | 20 (29.4) | 70 (37.4) | 188 (35.3) | |
| Not dedicated well | 50 (18.0) | 18 (26.5) | 41 (21.9) | 109 (20.5) | |
| **Total number of brothers and sisters** | | | | | 0.52 |
| No brothers/sisters | 27 (5.3) | 5 (3.7) | 16 (4.5) | 48 (4.8) | |
| 1 | 139 (27.2) | 37 (27.2) | 87 (24.6) | 263 (26.3) | |
| 2 | 105 (20.5) | 38 (27.9) | 87 (24.6) | 230 (23.0) | |
| 3 or more | 240 (47.0) | 56 (41.2) | 163 (46.2) | 459 (45.9) | |
| **Chronic disease status** | | | | | 0.77 |
| Diabetes | 53 (10.4) | 14 (10.3) | 29 (8.2) | 96 (9.6) | |
| Hypertension | 99 (19.4) | 23 (16.9) | 61 (17.3) | 183 (18.3) | |
| Chronic kidney disease | 3 (0.6) | 2 (1.5) | 3 (0.8) | 8 (0.8) | |
| No chronic disease | 356 (69.7) | 97 (71.3) | 260 (73.7) | 713 (71.3) | |
| **Family history of chronic or end-stage kidney disease** | | | | | 0.55 |
| Yes | 34 (6.7) | 8 (5.9) | 16 (4.5) | 58 (5.8) | |
| No | 423 (82.8) | 118 (86.8) | 302 (85.6) | 843 (84.3) | |
| Unknown | 54 (10.6) | 10 (7.4) | 35 (9.9) | 99 (9.9) | |
| **Health insurance enrollment types** | | | | | 0.7 |
| National Health Insurance (Community) | 138 (27.0) | 41 (30.1) | 101 (28.6) | 280 (28.0) | |
| National Health Insurance (Workplace) | 324 (63.4) | 83 (61.0) | 211 (59.8) | 618 (61.8) | |
| Full-aided | 12 (2.3) | 1 (0.7) | 6 (1.7) | 19 (1.9) | |
| Partial-aided | 4 (0.8) | 0 (0.0) | 2 (0.6) | 6 (0.6) | |
| Unsubscribed | 6 (1.2) | 2 (1.5) | 3 (0.8) | 11 (1.1) | |
| Unknown | 27 (5.3) | 9 (6.6) | 30 (8.5) | 66 (6.6) | |
| **Job** | | | | | 0.23 |
| Agriculture/forestry/fishing | 6 (1.2) | 1 (0.7) | 4 (1.1) | 11 (1.1) | |
| Self-employment | 32 (6.3) | 13 (9.6) | 14 (4.0) | 59 (5.9) | |
| Sales and service | 53 (10.4) | 10 (7.4) | 28 (7.9) | 91 (9.1) | |
| Production and labor | 49 (9.6) | 10 (7.4) | 39 (11.0) | 98 (9.8) | |
| Management/professional | 178 (34.8) | 52 (38.2) | 125 (35.4) | 355 (35.5) | |
| Housewives | 70 (13.7) | 21 (15.4) | 66 (18.7) | 157 (15.7) | |
| Students | 34 (6.7) | 12 (8.8) | 16 (4.5) | 62 (6.2) | |
| Unemployed/retired/others | 89 (17.4) | 17 (12.5) | 61 (17.3) | 167 (16.7) | |
| **Household monthly income** | | | | | 0.47 |
| Less than $1,600 | 94 (18.4) | 30 (22.1) | 69 (19.5) | 193 (19.3) | |

(*Continued*)

**Table 2.** (Continued)

| Characteristics | Willing to donate | Do not intend to donate | Undetermined | Total | p |
|---|---|---|---|---|---|
| | (N = 511) | (N = 136) | (N = 353) | (N = 1000) | |
| $1,600 to < $3,200 | 205 (40.1) | 52 (38.2) | 140 (39.7) | 397 (39.7) | |
| $3,200 to $4,800 | 128 (25.0) | 21 (15.4) | 78 (22.1) | 227 (22.7) | |
| More than $4,800 | 68 (13.3) | 30 (22.1) | 53 (15.0) | 151 (15.1) | |
| Unknown | 16 (3.1) | 3 (2.2) | 13 (3.7) | 32 (3.2) | |

[a]The values are expressed in number (%)

period after donor nephrectomy (66.0%), support for kidney-related tests of national health examination after donor nephrectomy (64.5%), and support for salary during a leave from work (55.7%). Finally, 73.2% of the respondents agreed that there was a need for socioeconomic support for living kidney donors, which increased to 81.3% after reading the explanation of donor nephrectomy and possible short-term and long-term complications (S2 Fig; $p < 0.001$, McNemar's test).

## Opinions related to discrimination after live kidney donation

In a survey on the possibility of social disadvantages due to donations, 253 (25.3%) of the participants thought they would be discriminated against or disadvantaged at work after donation, and the main reason for discrimination is sick leave that living donors might not need. In total, 536 (53.6%) respondents answered that there would be no disadvantages before reading the description regarding live kidney donations. The proportion of respondents who answered "there will be disadvantages" significantly increased from 25.3% to 29.2% when comparing before and after exposure to the description (S3 Fig).

## Discussion

Surveys of the general population are essential, considering that this group can be potential kidney donors and the basis for social consensus. The current survey provides the only published data on perceptions and attitudes toward living donor kidney transplantation in the Korean population. We found that more than half of the respondents were willing to donate to their loved ones who needed a kidney, and most of the participants had positive thoughts on socio-economic support for living kidney donors.

Irving et al. [15] reported that the significant factors that influence the decision to be an organ donor are religious beliefs, cultural aspects, family/relational ties, body integrity, interaction with the health system, and knowledge. Asia reports the lowest deceased donor organ donation rate in the world [16, 17]. Traditionally, since Korea belongs to the Confucian-influenced society, deceased donor transplantation is not immensely active because of the belief that body integrity, i.e., the body created by their parents, must be preserved [18, 19]. Additionally, the reasons why deceased donor transplantation in Korea is less than that of living donors as follows: 1) the leading causes of brain death, such as death from head trauma caused by traffic or cerebrovascular accident, have gradually decreased as the development of emergency treatment did not lead to death; and 2) the rapid transition to an aging society has led to an increase in the number of brain deaths in older patients with various underlying diseases, so organs that can be donated are often limited [18–21]. Apart from this, familial common good and mutual obligation are also strongly emphasized in many Asian countries; therefore, the number of living donor kidney transplants is significantly higher than that in other

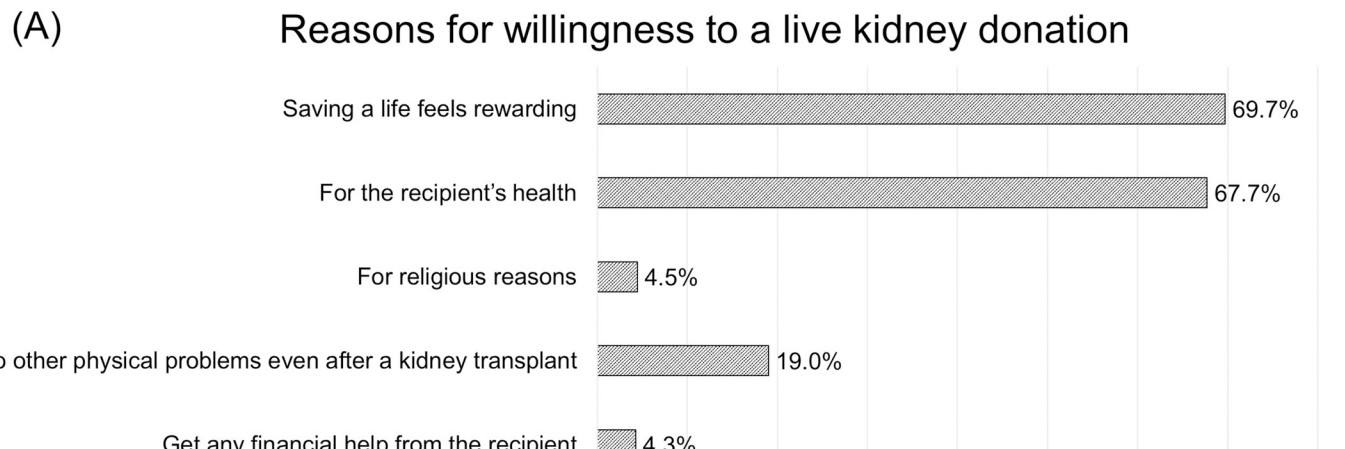

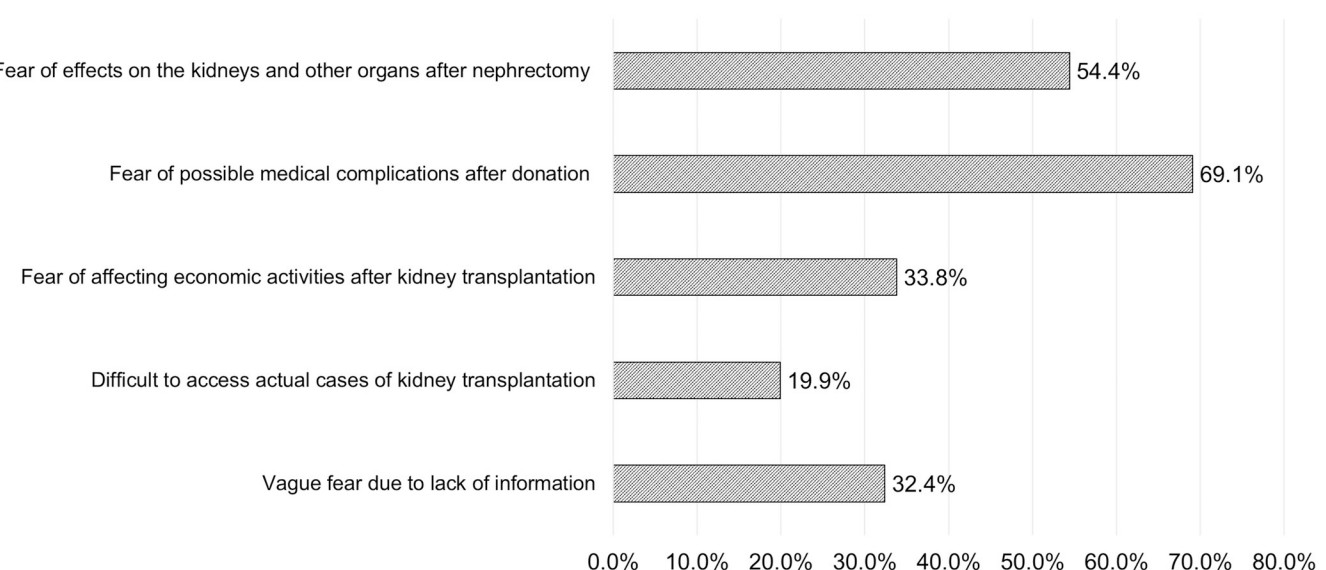

**Fig 2. Reasons for not donating a kidney according to a living kidney donation or not.** (A) Reasons for willingness to a live kidney donation. (B) Reasons for reluctance to a live kidney donation.

countries when considering the total number of transplantations, even without any special encouragement [21]. In this study, the willingness to donate a kidney was about half, and the targets of donations mainly were family members. There seems to be no great reluctance to donate a kidney to family members, probably because this may have originated from the

aforementioned Asian culture that values family relations. In this study, there was no significant difference between the willingness to donate kidneys and religious life.

Unlike in Western countries, live kidney donation is being actively conducted even though live kidney transplant is not systemically encouraged, which may be due to the family-centered culture in Korea. Ironically, the various disadvantages that live donors have after donor nephrectomy are not well known because the culture emphasizes the common good of the family community. The economic influence on donors is rarely considered but is understood by considering all possible direct and indirect expenses incurred, including travel for tests, appointments, hospital admission, incidental medical costs, lifelong regular health check-ups, and economic consequences of lost or impaired ability to work [22–24]. Nevertheless, donors still receive no benefits or social support in Korea. Reimbursement policies for donors may be a practical approach to alleviate kidney shortages [25]. Recent policies for reimbursement of the economic consequences incurred by donors have emerged mainly in Western countries, including the United States, United Kingdom, and Canada, rather than in Asian countries [22]. However, there is no national reimbursement policy for live donors in Korea.

One thing to note in this study is that more participants responded that they needed socioeconomic support after providing detailed information on live kidney donation. Even before reading the detailed information about live kidney transplants, 73.2% of respondents already agreed to economic/social support for donors, and after reading the description, this proportion increased to 81.3% (S2 Fig). We found that the general population judged that it is reasonable to appropriately compensate for the disadvantages that living kidney donors may experience after donor nephrectomy.

In particular, according to previous reports, lack of knowledge regarding organ donation and the process was usually reported as one of the main obstacles for transplantation [15]. Therefore, we tried to clarify whether the general population has the same attitudes toward live kidney donation, even after receiving accurate information. One of the main points of this study was to assess any changes in potential donors' attitudes towards live kidney donations when exposed to detailed information on donor nephrectomy and possible associated peri- and postoperative morbidities. Interestingly, the proportion of respondents who thought live kidney donation was safe increased by > 50% after reading the detailed explanation of the donation procedures, including adverse events. The low absolute numbers or proportions of surgery-related complications may allow the general population to consider donor nephrectomy more easily. However, they would not have occurred without kidney donation, which is only beneficial for the recipients and not for donors themselves. Therefore, it is necessary to emphasize that caution is essential even when minor medical complications are possible, and this must be included in the detailed explanation of live kidney donation before people decide whether to donate.

Some limitations of the present study should be noted. First, the survey was not validated or standardized. To the best of our knowledge, there are no adequate references in previous studies. Further research on the perception of the general population for live kidney donation should be performed to validate or standardize our survey. Additionally, the relatively small number of participants who responded to our questionnaire could introduce selection bias and limit generalizability, although we tried to balance the demographic factors of responders. In particular, since this survey was conducted via only e-mail, it is possible that individuals who were not familiar with electronic devices were excluded. However, age, sex, and the proportion of metropolitan residents of the total respondents did not differ significantly from the general population.

Since living donor kidney transplants are one of the best treatment alternatives to dialysis in patients with ESKD, the number of kidney donors will likely continue to increase. It is essential to continue to understand public perceptions and attitudes to promote "safe" and "impartial" live kidney donation. In this study, it was found that most of the general population had no objection to providing socioeconomic support to living kidney donors, although they recognized live kidney donation as relatively safe. Therefore, it is necessary to make every effort to make a safe and impartial live kidney donation without socioeconomic disadvantages to the donor based on social consensus.

## Supporting information

**S1 File. All questionnaires conducted in this study.**
(DOCX)

**S1 Fig. Level of awareness of living kidney donor transplants among the general population.**
(TIF)

**S2 Fig. Changes in the need for donor support before and after reading detailed kidney transplant descriptions: Whether the donor needs economic or social support.**
(TIF)

**S3 Fig. Changes in the need for donor support before and after reading detailed kidney transplant descriptions: Whether there is a possibility of a disadvantage after donation.**
(TIF)

**S1 Data.**
(XLSX)

## Author Contributions

**Conceptualization:** Eunjeong Kang, Hajeong Lee.

**Data curation:** Hajeong Lee.

**Formal analysis:** Eunjeong Kang, Jangwook Lee.

**Investigation:** Eunjeong Kang, Jangwook Lee, Yaerim Kim, Yong Chul Kim, Yon Su Kim, Hajeong Lee.

**Methodology:** Eunjeong Kang, Jangwook Lee, Sehoon Park, Yaerim Kim, Kwon Wook Joo, Insun Choi, Hajeong Lee.

**Project administration:** Insun Choi, Hajeong Lee.

**Resources:** Yong Chul Kim, Dong Ki Kim, Kwon Wook Joo, Yon Su Kim.

**Software:** Insun Choi.

**Supervision:** Jangwook Lee, Sehoon Park, Hyo Jeong Kim, Yong Chul Kim, Dong Ki Kim, Kwon Wook Joo, Yon Su Kim, Insun Choi, Hajeong Lee.

**Visualization:** Jangwook Lee.

**Writing – original draft:** Eunjeong Kang.

**Writing – review & editing:** Eunjeong Kang, Hajeong Lee.

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
