## [Decision Letter · Decision Letter 0]

5 Jul 2022

PONE-D-22-06811Perception regarding live kidney donation in the general population of South KoreaPLOS ONE

Dear Dr. Lee,

Thank you for submitting your manuscript to PLOS ONE. After careful consideration, we feel that it has merit but does not fully meet PLOS ONE’s publication criteria as it currently stands. Therefore, we invite you to submit a revised version of the manuscript that addresses the points raised during the review process.

We look forward to receiving your revised manuscript.

Kind regards,

P. Mangala CS De Silva

Academic Editor

PLOS ONE

Journal Requirements:

Reviewers' comments:

Reviewer's Responses to Questions

**Comments to the Author**

1. Is the manuscript technically sound, and do the data support the conclusions?

Reviewer #1: Yes

Reviewer #2: Partly

2. Has the statistical analysis been performed appropriately and rigorously? 

Reviewer #1: Yes

Reviewer #2: Yes

3. Have the authors made all data underlying the findings in their manuscript fully available?

Reviewer #1: No

Reviewer #2: Yes

4. Is the manuscript presented in an intelligible fashion and written in standard English?

Reviewer #1: Yes

Reviewer #2: Yes

5. Review Comments to the Author

Reviewer #1: Following errors suggested to be corrected.

Line 52: “…ranking second out of 70 countries in South Korea.” This is obviously an error. Is it Asia?

Line 53: “commitment “would be a more appropriate word than “devotion” in this context.

Line 74: “even in many donors” should be “even among many donors”.

Lines 182-185: “When multiple choices were allowed, participants wanted to donate their kidneys in the order of their offspring (86.3%), spouse (85.5%), sibling (80.2%), parent (75.9%), a close friend (31.9%), relative (19.0%), and others (6.8%)”. It would be better read as “When multiple choices were allowed, participants wanted to donate their kidneys to their offspring (86.3%), spouse (85.5%), sibling (80.2%), parent (75.9%), a close friend (31.9%), relative (19.0%), and others (6.8%), in that order.”

Lines 195 and 206: References in the text to the figures 3A and 3B are not clear. According the figures, it’s better to refer to the titles as “Reasons for willingness to a live-kidney donation” and “Reasons for reluctance to a live-kidney donation”.

Line 217: “leave of work” should be “leave from work”.

Line 251: This line gives the impression that deceased-donor kidney transplants (DDKT) are not done at all in Korea. But according to data, about 1500 DDKT were done in 2018.

Line 252: How can the high rate of deaths from head trauma be a reason for NOT doing DDKT?.

Line 264 and 265: “Unlike in Western countries, living kidney donation was not encouraged in Korea

because it was already performed based on a family-centered community culture in Korea”. This sentence is not clear. Please re-word it to give a proper meaning.

Table 2: First Column: Title should be “Willing to donate”.

Figures: There are two figs 2A. No Fig. 2B, although it’s referred to in the text.

Reviewer #2: This is a well written and very clear manuscript. The followings are my observations to improve the manuscript.

Key Words: Key words are not MeSH terms except Kidney transplantation. Please use MeSH terms

Study design

This study could have been planned as a qualitative study as a lot of information is restricted due to nature of the study design. Quantitative designing will not explore the exact picture of study in this nature.

Table 2

The first column – have a column title

the second column – Check the column title

P value – three decimal points may not be necessary. One or two decimal points would be adequate

% - Indicating percentage mark at the end of each value may not be required. You may include a foot note that the values are expressed in number (%). Then readers know that the value in the brackets is the percentage.

Figures

Many figures are a repetition of the content in the body. If the figure does not explain anything in addition to the content in the body, such figures can be taken out

6. PLOS authors have the option to publish the peer review history of their article (what does this mean?). If published, this will include your full peer review and any attached files.

Reviewer #1: No

Reviewer #2: No

---

## [Author Response · Author response to Decision Letter 0]

15 Jul 2022

Thank you for carefully reviewing this manuscript. Please refer to the "Response to Reviewers.docx" file submitted.

---

## [Editor Report · Decision Letter 1]

21 Jul 2022

Perception regarding live kidney donation in the general population of South Korea

PONE-D-22-06811R1

Dear Dr. Lee,

We’re pleased to inform you that your manuscript has been judged scientifically suitable for publication and will be formally accepted for publication once it meets all outstanding technical requirements.

Kind regards,

P. Mangala CS De Silva

Academic Editor

PLOS ONE
---

## [Editor Report · Acceptance letter]

26 Jul 2022

PONE-D-22-06811R1 

Perception regarding live kidney donation in the general population of South Korea 

Dear Dr. Lee:

I'm pleased to inform you that your manuscript has been deemed suitable for publication in PLOS ONE. Congratulations! Your manuscript is now with our production department. 

Kind regards, 

on behalf of

Professor P. Mangala CS De Silva 

Academic Editor

PLOS ONE